



# Observational evidence of EPP-NO$_x$ interaction with chlorine curbing Antarctic ozone loss

Emily M. Gordon[1], Annika Seppälä[1], Bernd Funke[2], Johanna Tamminen[3], and Kaley A. Walker[4]

[1]Department of Physics, University of Otago, Dunedin, New Zealand
[2]Instituto de Astrofísica de Andalucía, Granada, Spain
[3]Space and Earth Observation Centre, Finnish Meteorological Institute, Helsinki, Finland
[4]Department of Physics, University of Toronto, Toronto, Canada

**Correspondence:** Annika Seppälä (annika.seppala@otago.ac.nz)

**Abstract.**

We investigate the impact of the so-called energetic particle precipitation (EPP) indirect effect on lower stratospheric ozone, ClO and ClONO$_2$ in the Antarctic springtime. We use observations from Microwave Limb Sounder (MLS) and Ozone Monitoring Instrument (OMI) on Aura, Atmospheric Chemistry Experiment - Fourier Transform Spectrometer (ACE-FTS) on

SciSat, and Michelson Interferometer for Passive Atmospheric Sound (MIPAS) on Envisat, covering the overall period of 2005-2017. Using the A$_p$ index to proxy EPP, we find consistent ozone increases with elevated EPP during years with easterly phase of the quasi biennial oscillation (QBO) in both OMI and MLS observations. While these increases are opposite to what has been previously reported at higher altitudes, the pattern in the MLS O$_3$ follows the typical descent patterns of EPP-NO$_x$. The ozone enhancements are also present in the OMI total O$_3$ column observations. Analogous to the descent patterns found

in O$_3$, we also found consistent decreases in springtime MLS ClO following winters of elevated EPP. To verify if this is due to a previously proposed mechanism of conversion of ClO to the reservoir species ClONO$_2$ in reaction with NO$_2$, we used ClONO$_2$ observations from ACE-FTS and MIPAS. As ClO and NO$_2$ are both catalysts in ozone destruction, the conversion into ClONO$_2$ would result in ozone increase. We find a positive correlation between EPP and ClONO$_2$ in the upper stratosphere in the early spring, and the lower stratosphere in late spring, providing the first observational evidence supporting the

previously proposed mechanism relating to EPP-NO$_x$ modulating Cl$_x$ driven ozone loss. Our findings suggest that EPP has played an important role in modulating ozone depletion in the last 15 years. As chlorine loading in the polar stratosphere continues to decrease in the future, this buffering mechanism will become less effective and catalytic ozone destruction by EPP-NO$_x$ will likely become a major contributor to Antarctic ozone loss.

## 1  Introduction

Our understanding of the causes of the Antarctic stratospheric ozone hole (Farman et al., 1985), relies on half a century of discoveries about the Earth's atmosphere: The Brewer-Dobson circulation (Brewer, 1949), which allows gases such as chlorofluorocarbons (CFCs) emitted in the tropical troposphere to be drawn into the southern polar atmosphere; The strong polar vortex in the Southern Hemisphere, which allows the polar stratosphere to become very cold, with a net down-welling





effect pulling gases from the mesosphere and upper stratosphere into the lower stratosphere (Schoeberl and Hartmann, 1991);

Polar stratospheric clouds (PSCs), forming in the very cold lower stratosphere which, with the reintroduction of sunlight in the early spring, enable the breakdown of CFCs and chlorine reservoirs into simpler $Cl_x$ (= Cl + ClO) molecules on the cloud surfaces (Solomon et al., 1986). $Cl_x$ is effective at catalytically destroying ozone with the chain of reactions:

$$ClO + O \rightarrow Cl + O_2 \tag{R1}$$

$$Cl + O_3 \rightarrow ClO + O_2 \tag{R2}$$

$$Net : O + O_3 \rightarrow 2O_2$$

Other, more complicated heterogeneous reactions also destroy ozone (Brasseur and Solomon, 2005), but they will not be considered further here.

In the lower stratosphere, $Cl_x$ is, for most of the year, stored in reservoir species such as HCl and $ClONO_2$. $Cl_x$ is activated from these species in heterogeneous reactions in the springtime, hence the importance of PSCs providing solid and liquid particles. As PSCs disappear with warming of the stratosphere as spring progresses, $Cl_x$ is converted back to these reservoirs via reactions such as:

$$ClO + OH \rightarrow HCl + O_2 \tag{R3}$$


$$Cl + HO_2 \rightarrow HCl + O_2 \tag{R4}$$

$$ClO + NO_2 \xrightarrow{M} ClONO_2 \tag{R5}$$

Reactions R3-R5 require the presence of $HO_x$ (= OH + $HO_2$) or $NO_x$ (= NO + $NO_2$) gases. This indicates that the presence

of $HO_x$ and $NO_x$ gases in the lower and middle stratosphere plays a critical role as a limiter in $Cl_x$ driven $O_3$ loss, until the eventual removal of $Cl_x$ from the atmosphere via gravitational sedimentation of HCl.

In the context of polar ozone loss, in the past 20 years we have learned more about the impact of energetic particle precipitation (EPP). EPP is the flux of charged particles of solar and magnetospheric origin into the Earth's atmosphere. For the most part, this is made up of energetic electrons, with solar proton events (SPE, precipitation of energetic protons) being more

sporadic (Seppälä et al., 2014). Energetic particles ionise the neutral atmosphere, and the resulting chain of reactions is a key source of $NO_x$ and $HO_x$ in the mesosphere and upper stratosphere. $NO_x$ and $HO_x$ act as catalysts in ozone depleting reaction cycles such as:

$$NO_2 + O \rightarrow NO + O_2 \tag{R6}$$





$$NO + O_3 \rightarrow NO_2 + O_2 \tag{R7}$$

$$Net : O + O_3 \rightarrow 2O_2$$

and

$$HO_2 + O \rightarrow HO + O_2 \tag{R8}$$


$$HO + O_3 \rightarrow HO_2 + O_2 \tag{R9}$$

$$Net : O + O_3 \rightarrow 2O_2$$

and this EPP driven ozone loss has been the topic of a large number of studies in the past decades. Note here that this role of
$HO_x$ and $NO_x$ in ozone balance is opposite to their ozone loss limiting impact via the build up of reservoirs such as $ClONO_2$
in the lower and middle stratosphere.

The role of $HO_x$ and $NO_x$ in *in situ* EPP driven ozone loss is now well known (see e.g. Jackman et al., 2008; Andersson
et al., 2014). Polar ozone is also affected via the so called "EPP indirect effect" (Randall et al., 2006). This refers to the process
of transport of $NO_x$, produced by energetic particles at altitudes above 50 km ("EPP-$NO_x$"), to stratospheric altitudes, where it
can contribute to ozone loss. When EPP occurs over the winter poles, the lack of sunlight results in an increased photochemical
lifetime of $NO_x$, and the stable atmosphere provides a route for down-welling to stratospheric altitudes. This mechanism
for EPP-$NO_x$ descent is well documented (see e.g. Siskind et al., 2000; Randall et al., 2005; Seppälä et al., 2007; Randall
et al., 2007; Funke et al., 2014a, b; Gordon et al., 2020), and the depleting effect on ozone has been reported by a number of
studies: e.g. Randall et al. (2005) used observations from HALOE (HALogen Occultation Experiment), SAGE (Stratospheric
Aerosol and Gas Experiment) II and III, POAM (Polar Ozone and Aerosol Measurement) II and III, MIPAS (Michelson
Interferometer for Passive Atmospheric Sounding) and OSIRIS (Optical Spectrograph and InfraRed Imager System) to detect
the $NO_x$ increases in the Northern Hemisphere in January to March 2004 and reported ozone loss in March 2004 in the polar
stratosphere. This was attributed to the combination of geomagnetic activity occurring in the winter, and the reformation of the
polar vortex following a Sudden Stratospheric Warming (SSW) earlier in the winter. Seppälä et al. (2007) used the geomagnetic
index $A_p$ as a proxy for EPP levels. They correlated the 4-month wintertime average $A_p$ value with the wintertime $NO_2$ data
from GOMOS (Global Ozone Monitoring by Occultation of Stars) from 2002 to 2006 for both hemispheres, finding a robust
linear relationship between the two. They also note ozone loss and suggest it is due to the descent of EPP-$NO_x$. Damiani et al.
(2016) looked directly at ozone observations from the Solar Backscatter Ultraviolet Radiometer (SBUV) and the Microwave



Limb Sounder (MLS, on the Aura satellite), together spanning the period 1979-2014. They find ozone depletion of around 10-15% descending to 30 km (middle stratosphere) in September before disappearing. By comparing this with simultaneous $HNO_3$ enhancements in the Aura period (2004-2014), they were able to attribute the ozone depletion to $NO_x$ increases from EPP ($HNO_3$ is a reservoir of $NO_x$).

Recent studies have looked at the descent of $NO_x$ in the Southern Hemisphere in more detail. Funke et al. (2014a) used tracer correlations to extract EPP-$NO_y$ ($NO_y$ = all reactive nitrogen) from total $NO_y$, and found it reaching altitudes as low as 20-25 km in the Southern Hemisphere by September. In the Antarctic spring, these correspond to altitudes where the ozone hole forms. Gordon et al. (2020) use a similar $A_p$ scheme to Siskind et al. (2000) and Seppälä et al. (2007) to detect EPP-$NO_2$ in the stratospheric total $NO_2$ column using observations from the Ozone Monitoring Instrument (OMI). They find that the $NO_2$ column is significantly correlated with $A_p$ until November. This presence in the $NO_2$ stratospheric column suggests perturbations in EPP-$NO_2$ contribute significantly to the overall amount of $NO_2$ present in the stratosphere, as well as indicating that the EPP-$NO_y$ reported by Funke et al. (2014a) remains in the atmosphere longer, until the breakdown of the polar vortex. Gordon et al. (2020) also found that accounting for the phase of the Quasi-Biennial Oscillation (QBO) results in increased correlation between $A_p$ and the stratospheric $NO_2$ column in years with easterly phase of the QBO, and opposite for the westerly QBO phase. They postulate that this modulation by the QBO could reflect the influence of the QBO on the primary (non-EPP) $NO_x$ source via transport from the equatorial region, combined with the effect the QBO has on polar temperatures, which would influence the efficiency of removal of nitrogen species from the polar stratosphere.

## 1.1 This work

Here, we investigate the effect of EPP-$NO_x$ on stratospheric ozone focusing on the time of the ozone hole formation in the spring. Our analysis follows on from the results reported by Gordon et al. (2020), now focusing on the implications of the enhanced stratospheric $NO_2$ column on Antarctic stratospheric ozone balance. We use ozone and chlorine species observations from three different satellite platforms (and four instruments), spanning the time period of 2005-2017, to get a more cohesive view on interactions taking place with EPP-$NO_x$, atmospheric chlorine, and ozone. We control our analysis for EPP levels (as proxied by the $A_p$ index) and the phase of the QBO. Following from the initial analysis of ozone, we examine how EPP affects $Cl_x$ activation in the springtime, by using ClO observations from MLS, and $ClONO_2$ from Atmospheric Chemistry Experiment - Fourier Transform Spectrometer (ACE-FTS) and MIPAS observations. We find that ozone tends to increase in years with high EPP and easterly QBO, and suggest that this could be attributed to the combined effect of EPP and the QBO on the activation, and deactivation of $Cl_x$.



## 2   Observations and Methodology

### 2.1   MLS

We use ozone and ClO profiles (v4.2) from the Microwave Limb Sounder (MLS), on the Aura satellite (Schwartz et al.,
115   2015; Santee et al., 2015). The data has been sorted according to Livesey et al. (2017), i.e removing data that do not meet the
recommended quality standards. The $O_3$ profiles have been validated by Froidevaux et al. (2008), with further comparison to
ground-based and other satellite measurements by Hubert et al. (2016). Here, we use stratospheric $O_3$ observations (15 km to
50 km) with vertical resolution around 3 km, and uncertainty of no more than 4%.

MLS ClO is valid throughout the stratosphere although the lower-most altitudes (15-18 km) suffer from a negative bias.
The bias, which has been uniform throughout the MLS period, is least significant in the polar region and is also systematic:
Each latitude is affected the same way. We mitigate the effect of the bias by looking at anomalies as any systematic bias will
not affect the overall gradient of the trend. Since anomalies are differences from a mean, any shift is cancelled in subtraction.
The vertical resolution of stratospheric ClO is around 3 km and the error on individual profiles is around $\pm 0.1$ ppbv (Livesey
et al., 2017). We do not use ClO from dusk until dawn (i.e. nighttime) due to rapid conversion of ClO to the $Cl_2O_2$ dimer at
nighttime (Brasseur and Solomon, 2005). Excluding these measurements avoids the change in partitioning between day and
night. We sort for day by only using profiles with solar zenith angle $< 90°$. MLS ClO profiles have been validated by Santee
et al. (2008).

### 2.2   OMI

We analyse ozone total column data from the Dutch-Finnish built Ozone Monitoring Instrument (OMI), also on Aura (Bhartia,
2012). Here we use the OMI $O_3$ version 3, level 2 daily gridded product ($0.25° \times 0.25°$ OMTO3G version 3). The algorithm
is described by Bhartia (2002) and Bhartia (2007) with validation of OMI $O_3$ reported by McPeters et al. (2008). OMI total
$O_3$ column measurements have an estimated error of around 1-2%. The ozone column is provided in DU. Since 2007, OMI
has been experiencing an issue known as the row anomaly, where certain fields of view are blocked (Schenkeveld et al., 2017).
This issue has been accounted for in the data used here, and we exclude all row anomaly affected data in this study.

### 2.3   ACE-FTS

Atmospheric Chemistry Experiment – Fourier Transform Spectrometer (ACE-FTS) is an instrument on the Canadian SciSat
satellite (see e.g. Boone et al., 2005). We use ACE $ClONO_2$ level 2, version 4.0 sorted according to Boone et al. (2019),
removing recommended outliers (Sheese et al., 2015). We use only profiles in the Southern polar region (poleward of 60S) for
the months August and September. Like MLS ClO, negative bias exists in $ClONO_2$ but, as for MLS ClO, this is mitigated
here through the use of anomaly studies as though the bias is altitude dependent, it is consistent in time throughout the data set.
ACE $ClONO_2$ has been validated by Wolff et al. (2008) and more recently by Sheese et al. (2016).





## 2.4 MIPAS

Michelson Interferometer for Passive Atmospheric Sounding (MIPAS) is a limb sounding instrument on the European Space Agency's Envisat-satellite. Here we use the Institut für Meteorologie und Klimaforschung (IMK) at Forschungszentrum Karl-sruhe and the Instituto de Astrofísica de Andalucía (IAA) product. The algorithm is described by von Clarmann et al. (2009)). MIPAS was fully operational from July 2002 until March 2004. An error with the instrument then resulted in reduced duty cycles and data holes, with full coverage resuming in January 2006, lasting until February 2012. Here, we use MIPAS $ClONO_2$ observations (V5R_CLONO2_222/223) for the Antarctic springtime from years 2006-2011. We exclude 2002 due to the sud-den stratospheric warming that occurred in the SH that spring, disrupting the polar stratosphere therefore any $NO_x$ descent. We also exclude November 2003 due to the extremely large SPEs, known as the Halloween event, that occurred throughout late October and early November of that year. These events caused large amounts of particle precipitation resulting in *in situ* $NO_x$ increases in the Antarctic stratosphere (López-Puertas et al., 2005). These increases in November would likely mask any EPP effects from the previous winter. The 2004 and 2005 springs are not included due to the aforementioned instrumental anomaly. MIPAS $ClONO_2$ observations have been validated by Höpfner et al. (2007), and were found to be consistent with ACE-FTS $ClONO_2$ by both Wolff et al. (2008) and Sheese et al. (2016).

## 2.5 EPP Proxy

Analogous to Gordon et al. (2020), we use the geomagnetic activity index $A_p$ as a proxy for the overall winter EPP levels. We take the mean $A_p$ index from May to August of each individual year (consistent with previous studies of e.g. Siskind et al. (2000); Seppälä et al. (2007)) and denote this 4-month mean $A_p$ as $\hat{A}_p$. The average $\hat{A}_p$ for the study period was 8.3 and the $\hat{A}_p$ values for each individual year are given in Table 1.

## 2.6 QBO

To account for the influence of the QBO in our analysis (see Gordon et al., 2020), we bin the years according to the phase of the QBO in May. To determine the phase of the QBO, we use the equatorial zonal mean zonal wind at the 25 hPa level (see Baldwin and Dunkerton, 1998, for explanation of use of this level in the SH). Years where the zonal mean zonal wind is easterly are designated easterly QBO (eQBO), while westerly winds are designated westerly QBO (wQBO). The QBO phase for each year of the study is listed in Table 1.

## 2.7 Methods: Anomalies and Correlation

We analyse correlation between $\hat{A}_p$ and various trace gases in the atmosphere. For this purpose, we use the Spearman rank correlation coefficient $\rho$, which correlates two non-normally distributed datasets (von Storch and Zwiers, 1999). For signifi-cance testing purposes, the correlation is characterised as significant if the $p$-value is less than 0.05, that is, the correlation is significant at 95% or higher.





**Table 1.** The average $A_p$ from May to August ($\hat{A}_p \pm 2\times$ standard error in the mean), designation to high or low $A_p$ group ("h-$\hat{A}_p$" for high $A_p$, "l-$\hat{A}_p$" for low $A_p$), and the phase of the QBO in May for each of the years included in the analysis.

| Year | $\hat{A}_p$ | | QBO |
|------|-------------|--------------|-----|
| 2005 | $13.9 \pm 2.9$ | h-$\hat{A}_p$ | E |
| 2006 | $7.6 \pm 1.2$ | l-$\hat{A}_p$ | W |
| 2007 | $6.8 \pm 1.0$ | l-$\hat{A}_p$ | E |
| 2008 | $5.8 \pm 0.7$ | l-$\hat{A}_p$ | W |
| 2009 | $4.3 \pm 0.6$ | l-$\hat{A}_p$ | E |
| 2010 | $6.9 \pm 1.3$ | l-$\hat{A}_p$ | E |
| 2011 | $8.1 \pm 1.3$ | l-$\hat{A}_p$ | W |
| 2012 | $9.5 \pm 1.8$ | h-$\hat{A}_p$ | E |
| 2013 | $10.0 \pm 1.6$ | h-$\hat{A}_p$ | W |
| 2014 | $6.2 \pm 0.9$ | l-$\hat{A}_p$ | E |
| 2015 | $11.1 \pm 2.1$ | h-$\hat{A}_p$ | W |
| 2016 | $9.7 \pm 1.5$ | h-$\hat{A}_p$ | W |
| 2017 | $8.4 \pm 1.4$ | h-$\hat{A}_p$ | E |

Correlation studies can be misleading in their results as they view data through a purely statistical lens and do not account for underlying physics. Here, significance of a correlation is tested if we have a reason to speculate on a connection based on known physical or chemical properties or analysis of observational data. Thus, we first check for evidence in anomalies of observational data. As discussed in the Introduction, work by Gordon et al. (2020) has shown evidence that EPP (as proxied by $\hat{A}_p$) and QBO affect trace gases in the stratosphere. Here, we will examine the composite anomalies for different combinations of QBO phase and $\hat{A}_p$ level for each trace gas analysed. Years with $\hat{A}_p > 8.3$ are designated as high $\hat{A}_p$ (h-$\hat{A}_p$) and those with $\hat{A}_p < 8.3$ are designated as low $\hat{A}_p$ (l-$\hat{A}_p$). 8.3 is chosen as it is the mean $\hat{A}_p$ for the study period. These are indicated in Table 1.

In the time period under investigation there has been a reduction in equivalent effective stratospheric chlorine (EESC). This reduction in chlorine and the following gradual recovery of stratospheric ozone has been mitigated in the analysis by de-trending the observations for all correlation calculations. Here, detrending was performed by calculating the gradient of the yearly trend with a linear least squares fit, then subtracting this from the data. This was not applied to the results presenting composite anomalies, which are shown here as an indication of the overall variability in the volume mixing ratios.

We note that other factors can also pay a role in Antarctic stratospheric ozone levels, most notably solar spectral irradiance (SSI) varying with the 11-year solar cycle, and the El Niño–Southern Oscillation (ENSO). Due to the limited time series of observations, it is not possible to robustly control for all. However, we note that the effect of SSI has limited influence on springtime Antarctic ozone variability and the effects are mainly limited to above 10 hPa level (Matthes et al., 2017). Some studies have suggested that ENSO can both influence stratospheric ozone variability (Lin and Qian, 2019), and potentially be





influenced by Antarctic ozone variability (Manatsa and Mukwada, 2017). But, as with solar irradiance, the ENSO influence on

Antarctic ozone variability appears to be limited to the upper stratosphere, above the 10 hPa level (Lin and Qian, 2019).

## 3 Indirect effect on springtime Antarctic ozone

### 3.1 MLS profile observations

To find the indirect effect of EPP on ozone in the springtime Antarctic stratosphere, we analyse the ozone anomaly for 4

different categories: high $\hat{A}_p$ & eQBO, low $\hat{A}_p$ & eQBO, high $\hat{A}_p$ & wQBO, and low $\hat{A}_p$ & wQBO (see Table 1). The average

MLS polar (60°S-82°S) $O_3$ from 2005 to 2017 is shown in Figure 1a) as a composite zonal 3-day running mean. Hereafter, all

data averaged over a range of polar latitudes are area weighted by *cos(latitude)* to avoid emphasising the highest latitudes. The

vertical axis is pressure from 100 hPa (approx. 18 km) to 1 hPa (approx. 50 km), while the horizontal axis is time from early

August until the end of December. Here, we can see the ozone hole forming at pressure levels below 20 hPa (altitude $<\sim$28 km)

from September. Panels b)-e) show the composite anomaly from the mean (panel a) for the 4 different combinations of $\hat{A}_p$ and

QBO phase. Years with high $\hat{A}_p$ (panels b and d) exhibit a positive anomaly of around +0.1 ppmv increased in ozone in the

middle stratosphere in August and September ($\sim$20 hPa), while low $\hat{A}_p$ years (panels c and e) show the opposite (reduced

ozone). This implies that positive anomaly in the middle stratosphere in August and September could be linked to high $\hat{A}_p$.

Years with eQBO (b and c) display a positive anomaly ($\sim$+0.1 ppmv or $<$10 % reduction from the mean) in the middle

stratosphere in October, while wQBO years (d and e) show the opposite. This suggest the anomaly is likely related to the QBO

phase and could be linked to the effect noted by Garcia and Solomon (1987) and Lait et al. (1989): more ozone is present in the

Southern polar stratosphere in years with eQBO. In the lower stratosphere in November, positive (negative) anomaly occurs

in high (low) $\hat{A}_p$ years. This indicates that these changes are linked to EPP: high $\hat{A}_p$ results in ozone increases in November.

In December, in the middle stratosphere ($\sim$20 hPa) high $\hat{A}_p$ appears to results in negative ozone anomaly ($\sim -0.1$ ppmv or

$< 10$ % reduction from the mean).

The above analysis indicates increases in ozone associated with high $\hat{A}_p$, and thus high EPP, while also finding ozone

decreases associated with the westerly phase of the QBO. We now look to see if the ozone increases linked to $\hat{A}_p$ are correlated

with $\hat{A}_p$ levels, and how this is modulated by the QBO phase. This is presented in Figure 2 for a) all years, b) eQBO years,

and c) wQBO years. As in Figure 1, ozone is *cos(latitude)* weighted zonal mean average over 60°S to 82°S. Note that for

all correlation analyses presented here, the data has been linearly detrended to avoid misattribution from linear increases or

decreases from reduced EESC since 2005. There is significant anti-correlation ($\rho \sim -0.4$ to $-0.6$) in the upper stratosphere

around 2 hPa in panels a) and b). This suggests that increases in $\hat{A}_p$ indeed result in ozone loss in this area, particularly during

eQBO. These ozone reductions are consistent with $O_3$ loss due to the EPP-$NO_x$ descending in the polar vortex. However, in

panel b), for eQBO conditions, the negative correlation pattern, which descends in time, is accompanied by a strong positive

correlation ($\rho > 0.6$) below $\sim$10 hPa in November. This indicates that EPP in eQBO years also contributes to ozone increases.

At this time both panels a) and b) show positive correlation in the middle and lower stratosphere, though this is only statistically

significant during eQBO years. These results seem to suggest that increased $\hat{A}_p$ results in ozone enhancement in November,


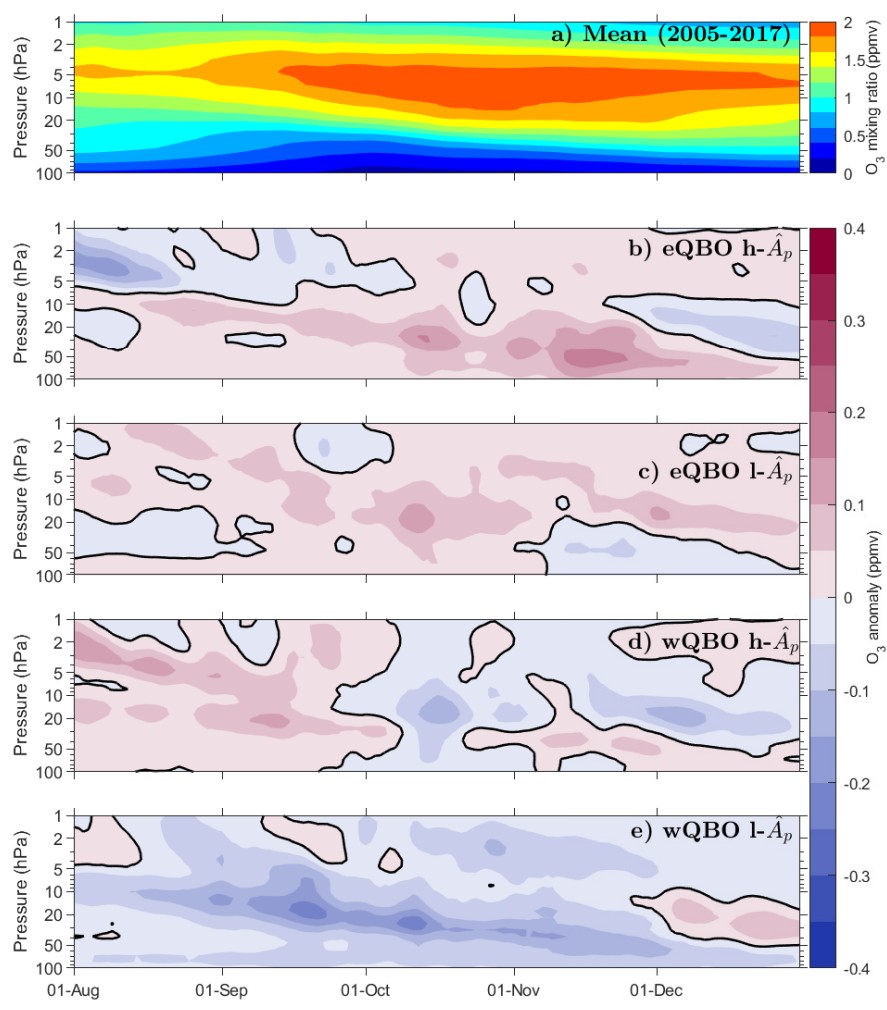

**Figure 1.** MLS profile ozone: a) Composite zonal mean polar ozone (60°S - 82°S) mixing ratio for the study period (2005–2017) from early August until December 31st. Horizontal axis is date and vertical axis is pressure from 100 hPa (∼18 km) to 1 hPa (∼50 km). Contour interval is 0.2 ppmv. b) Anomaly from the mean for years with high $\hat{A}_p$ and eQBO. Contour interval 0.05 ppmv with black line indicating the zero contour. Axes as above. c)-e) as b) but for different combinations of $\hat{A}_p$ and QBO phase (see individual panels). All data has been weighted by *cos(latitude)*.

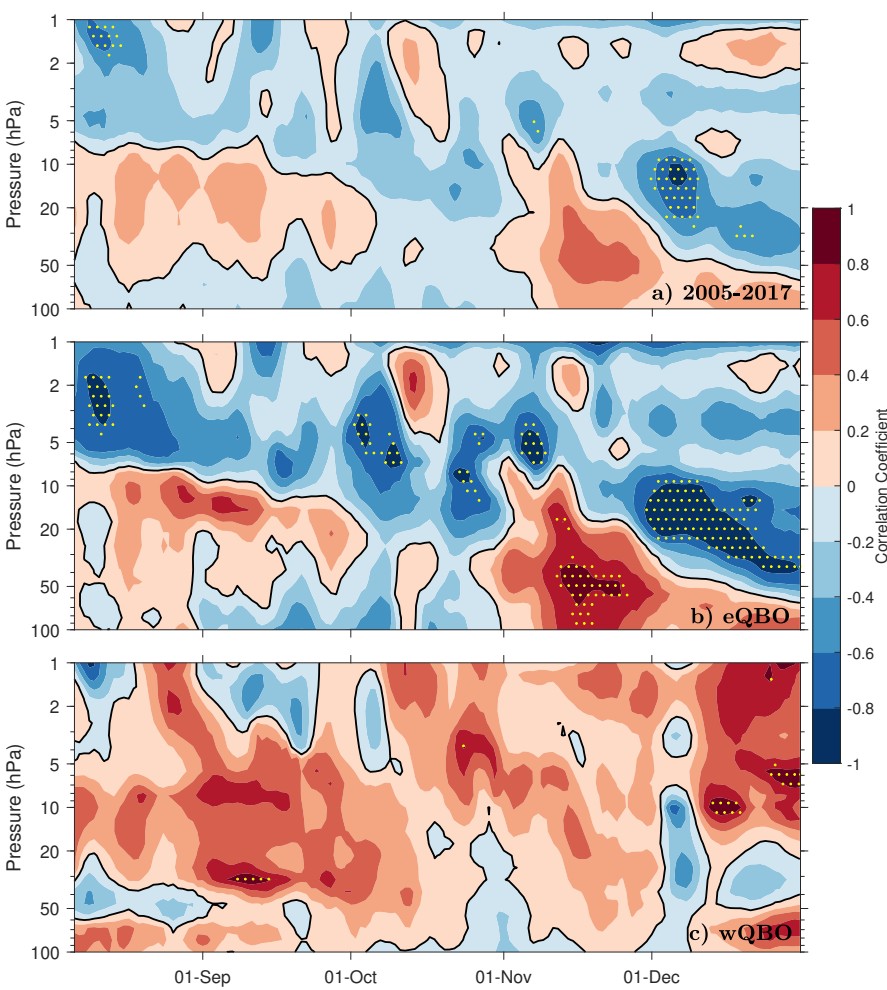

**Figure 2.** Correlation between area weighted polar ($60°$S to $82°$S) MLS ozone mixing ratio and $\hat{A}_p$ for a) all years of the study, b) eQBO years and c) wQBO years. Vertical axis is pressure in hPa, horizontal axis is time from the beginning of August until the end of December. Contours show the correlation coefficient with 0.2 interval (black contour for zero) and stippling indicates statistical significance ($p < 0.05$).





and that eQBO strengthens this relationship. There is little consistent correlation present in panel c) – there is no clear relation between polar springtime ozone profile variability and $\hat{A}_p$ in wQBO years.

## 3.2 OMI column observations

We now repeat the analysis for daily OMI total ozone column, instead of profile measurements. This is to verify whether the changes in ozone associated with EPP and the QBO are detectable in the ozone total column. While we lose the information contained in vertical profiles, we gain higher horizontal resolution. Note that here the OMI data with 0.25° gridding has been averaged over 1° latitude bins.

The composite 3 day running mean $O_3$ column from 2005 to 2017 is presented in Figure 3a). The figure shows zonal mean
ozone for each latitude poleward of 50°S in 1° bins as spring progresses from early August to the end of the December. Note that 1) no area weighting is required here, and 2) there are no data for the polar night as OMI $O_3$ is measured with back-scattered solar radiation. A key feature in this panel is the formation of the ozone hole in the springtime, with minimum values in ozone of less than 150 DU in late September-early October at the pole. Panels b)-e) show the composite anomaly from the mean (panel a) for the same combinations of QBO phase and $\hat{A}_p$ as before. Panel b) corresponds to the anomaly for eQBO and
h-$\hat{A}_p$ years. In this case the anomaly is almost entirely positive, with the largest values ($> 80$ DU) occurring in mid-November. This implies that the combination of high $\hat{A}_p$ and eQBO results in increased ozone throughout the springtime but especially in November. Panel c) (eQBO, low $\hat{A}_p$) is slightly more variable, especially in early spring. Easterly QBO appears to drive a positive ozone anomaly in October (as this appears in both eQBO panels), however, the sign of the anomaly changes in November, which seems to imply that low $\hat{A}_p$ results in $O_3$ decreases (up to $\sim -50$ DU) in November. Panel d) (wQBO, high
$\hat{A}_p$) is again variable throughout early spring, with positive anomalies mainly present at highest polar latitudes. As panel b) for eQBO, the positive anomaly (up to $\sim +50$ DU) in November for wQBO may be an indication that high $\hat{A}_p$ is linked to ozone increases at this time. Lastly, panel e) (wQBO, low $\hat{A}_p$) shows consistent negative anomaly: low $\hat{A}_p$ in wQBO years results in anomalously low ozone column ($\sim -40 - 50$ DU) throughout spring. These column ozone results are consistent with the MLS ozone profile anomalies below 20 hPa (Figure 1): In October and November the combination of high $\hat{A}_p$ and eQBO results in
anomalously high ozone, while low $\hat{A}_p$ and wQBO results in anomalously low ozone.

We now examine the correlation between ozone column (detrended) and $\hat{A}_p$ level. This is shown in Figure 4, with the panels from top to bottom presenting: a) all years, b) eQBO, and c) wQBO. Figure 4a) displays the correlation for all years of the study. Overall, the correlation $|\rho| < 0.6$ everywhere, with little statistical significance, when all years are taken into account and no QBO based binning is done. In panel b), for eQBO years, correlation is positive poleward of 60°S for almost all of spring.
Areas of significant positive correlation ($\rho \geq 0.6$) occur throughout August to October, and early November shows consistent significant positive correlation. This agrees with Figure 3: elevated $\hat{A}_p$ results in ozone increases at high Southern latitudes and this is more prevalent in eQBO years. At lower latitudes, between 50°S and 60°S there are patches of significant negative correlation. For wQBO years, shown in panel c), the correlation is highly variable, with $|\rho| < 0.4$, and not significant. Any influence of EPP on the ozone column is generally weaker during wQBO years. This is consistent with Gordon et al. (2020)
who reported significant correlation between stratospheric $NO_2$ column and EPP (as proxied by $A_p$) during eQBO years.



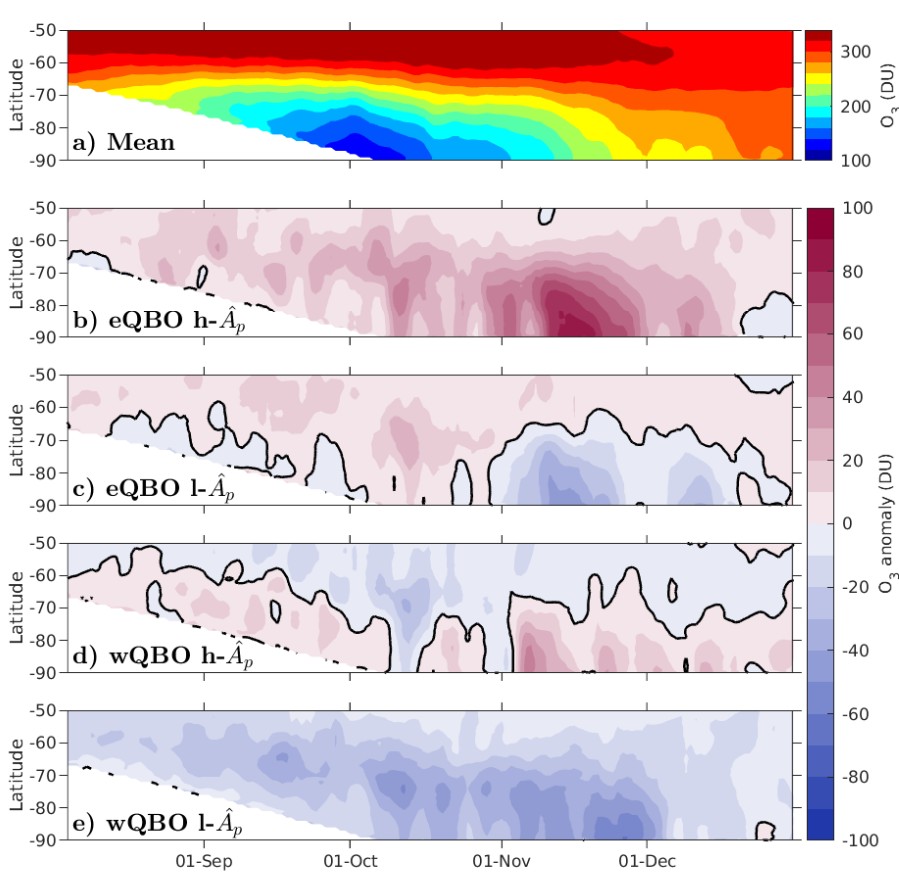

**Figure 3.** OMI ozone column: a) August-December 3 d running mean zonal mean column ozone from 2005 to 2017 for latitudes $50°$S-$90°$S in $1°$ bins (contour interval is 25 DU). b) Composite anomaly from the mean in panel a) for eQBO years with high $\hat{A}_p$. Horizontal and vertical axes as in a) with contour interval of 10 DU and 0-contour in black. c)-e) as b) but for different combinations of QBO phase and $\hat{A}_p$, see panel titles.

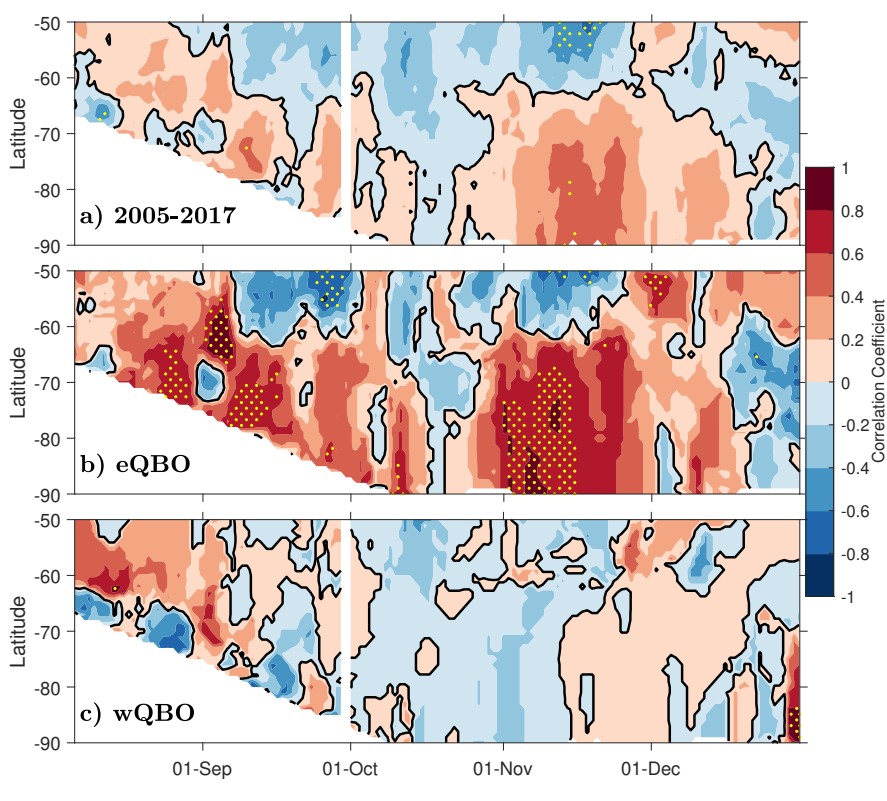

**Figure 4.** Correlation of zonal mean OMI $O_3$ and $\hat{A}_p$ from August to December at latitudes $50°$S to $90°$S for a) all years, b) eQBO years, and c) wQBO years. Contours represent correlation coefficient with contour interval of 0.2 (black line for zero level). Stippling indicates statistical significance at 95%.

To quantify the effect that enhanced EPP has on the total ozone column in the Southern Hemisphere spring, Figure 5a) presents the average polar ($60°$S to $90°$S) $O_3$ column in November as a function of $\hat{A}_p$ for the previous winter. Note here the ozone is both *cos(latitude)* area weighted, and detrended, to account for reduced EESC. Red triangles represent eQBO years and blue circles indicate wQBO. The figure also shows best-fit lines, with red fitting eQBO years, blue fitting wQBO, and yellow fitting all data. There is a robust linear relationship between $\hat{A}_p$ and ozone in eQBO years with the linear fit indicating an increase in November total ozone column by $1.4\,\mathrm{DU}/\hat{A}_p$, i.e. a 1.4 DU increase in the area weighted $O_3$ per unit increase in $\hat{A}_p$. The year-to-year variability for eQBO is in the range of 55-75 DU: $1.4\,\mathrm{DU}/\hat{A}_p$ would correspond to about 2 % change in ozone per unit increase in $\hat{A}_p$.

Figure 5a) accounted for the polar average with average $1.4\,\mathrm{DU}/\hat{A}_p$. Figure 5b) now shows this gradient of the linear fit between $\hat{A}_p$ and OMI ozone column in eQBO years for all points in the polar stratosphere with $1°$ latitude resolution. Stippling is taken from Figure 4b). We find that gradient of the fit is positive throughout November poleward of $60°$S. The





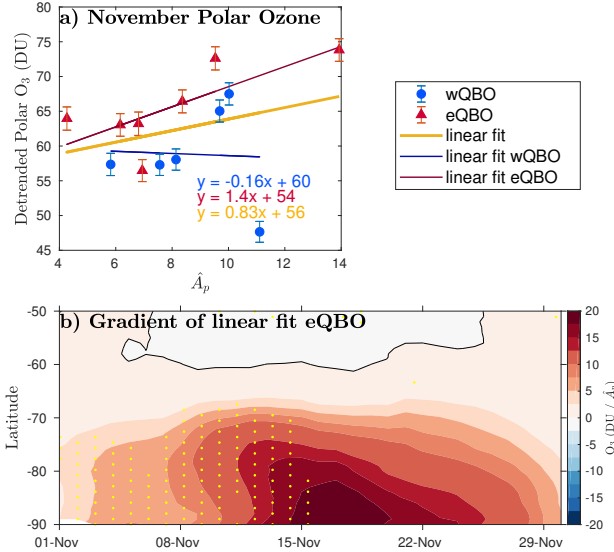

**Figure 5.** a) $\hat{A}_p$ vs. OMI detrended polar ozone averaged over $60°$S to $90°$S with *cos(latitude)* weighting. Blue circles indicate years with wQBO phase and red triangles indicate with eQBO. The lines show linear best fit with yellow fitting all data, blue fitting wQBO and red fitting eQBO. Corresponding equations for these are included in corresponding colours. Error bars are 2 times the standard error in the mean. b) Evolution of the gradient of the linear best fit between $\hat{A}_p$ and OMI $O_3$ over November in the polar stratosphere in eQBO years. The horizontal axis is time from the 1st to 30th of November and the y-axis is latitude from $90°$S to $50°$S. Contour interval is 2.5 DU/$\hat{A}_p$. Stippling from Figure 4b) is superimposed as reference to show where the OMI $O_3$ and $\hat{A}_p$ correlation was found to be significant at 95 % or higher. Recall eQBO years are [2005 2007 2009 2010 2012 2014 2017].

maximum contribution of $\hat{A}_p$ to column ozone occurs in mid November poleward of $80°$S, with increases of greater than 15 DU/$\hat{A}_p$, e.g. up to 15 DU increase in ozone south of $80°$S in mid November per unit increase in $\hat{A}_p$. These contributions occur simultaneously with significant correlation between the OMI $O_3$ column and $\hat{A}_p$ in early to mid November.

## 4   EPP indirect effect via chlorine species?

Our results indicate ozone increases, both below 20 hPa in profile observations and in the total ozone column, with enhanced EPP. Traditionally, the long term EPP effect on ozone has been considered to dominate via increased catalytic loss in $NO_x$ cycles. Earlier works of Jackman et al. (2000) and Funke et al. (2014a) have, however, suggested there may be a more complex interplay, with $NO_x$ interfering with ozone loss driving halogen species ClO and BrO. To our knowledge, this effect has not been previously verified from observations.

Funke et al. (2014a) showed, using MIPAS observations, that in the Antarctic stratosphere, EPP-$NO_y$ reaches altitudes as low as 22-25 km by September. They speculate on the effect this EPP-$NO_y$ might have on stratospheric ozone later in the spring,





suggesting that EPP-$NO_y$ could interfere with the buffering between ClO and $ClONO_2$ (via reaction R5, that is changing the partitioning between ClO and $ClONO_2$ by conversion of ClO to the inactive $ClONO_2$), and that "such EPP-induced

buffering of ClO could even outweigh the ozone loss by EPP-$NO_x$, resulting in a net reduction of the Antarctic chemical ozone loss" (Funke et al., 2014a). Gordon et al. (2020) provided additional evidence for the sustained descent of EPP-$NO_x$, further suggesting that the phase of the QBO during the winter months plays a role in $NO_x$ descent. Here, we found ozone increases under the same conditions. Motivated by the hypothesis of Funke et al. (2014a), we will now explore the mechanism they proposed: that EPP-$NO_x$ modulates the amount of active chlorine in the springtime, but also account for the phase of the

QBO.

## 4.1 MLS ClO observations

First, we investigate the composite mean ClO, and ClO anomaly, from MLS observations. This is done for years with different combinations of $\hat{A}_p$ and QBO phase as before and is presented in Figure 6. Panel a) illustrates the composite mean ClO averaged over 60°S to 82°S. Large amounts of ClO are activated in the lower stratosphere in the early spring (50 hPa – 20 hPa,

August through late September). This is followed by a large reduction in ClO mixing ratio, due to deactivation of chlorine with the reformation of its reservoirs (von Clarmann, 2013). Note that values below zero are a result of the known negative bias in MLS ClO. The split panels b) to e) show the composite anomaly for different combinations of $\hat{A}_p$ and QBO phase. As the abundance of ClO in the stratosphere changes dramatically throughout spring, each panel is divided into two, with the corresponding scale for each half indicated by the colour bar at the bottom of the columns. Note how in all panels, the

anomaly (regardless of the sign) appears to go from being contained in the upper or middle stratosphere in early spring (i.e. the left column), to a signal propagating down into the lower stratosphere in late spring (the right column). The descending anomalies are opposite for high and low levels of $\hat{A}_p$: the ClO anomaly is negative/positive for high/low $\hat{A}_p$ levels. Hence, high $\hat{A}_p$ years with negative ClO anomaly would indicate that EPP is associated with ClO decreases ($-0.0050$ to $-0.0125$ ppbv). This supports the above hypothesis that in years with high $\hat{A}_p$, and therefore more EPP-$NO_x$, we should find reduced ClO, as

enhanced $NO_2$ drives ClO to its $ClONO_2$ reservoir. The downward propagating signal closely resembles the typical descent pattern of EPP-$NO_x$ (see e.g. Funke et al., 2014a). Looking earlier in the season (left column), the descending anomalies can be traced up to 5 hPa level. In the lower stratosphere in early spring, the anomalies in general appear to be more linked to the phase of the QBO with eQBO/wQBO conditions leading to reduction/enhancement of ClO ($\sim \mp 0.02$ ppbv).

The correlation of MLS ClO and $\hat{A}_p$ is presented in Figure 7, in the same format as Figure 2. Panels a) (all years) and b)

(eQBO years) now show a similar downward propagating anti-correlation, starting from about 2 hPa in the beginning of August and reaching almost 50 hPa by November. This again agrees well with downward descent patterns of EPP-$NO_x$, known to be occurring at this time (see e.g. Funke et al., 2014a; Gordon et al., 2020). ClO being anti-correlated with EPP-$NO_x$ aligns with the hypothesis that EPP-$NO_x$ acts to drive ClO to its reservoirs. Our results show that this is more prevalent in eQBO years ($\rho \leq -0.8$ with $p \leq 0.05$), with wQBO years showing little significance. We also find a small positive region of significant

correlation in the lower stratosphere ($\sim 70$ hPa) in August. It is unlikely that any EPP-$NO_x$ has descended to such altitudes at this time. This could be related to some other mechanism, but won't be investigated further here. Panel c) (wQBO years)

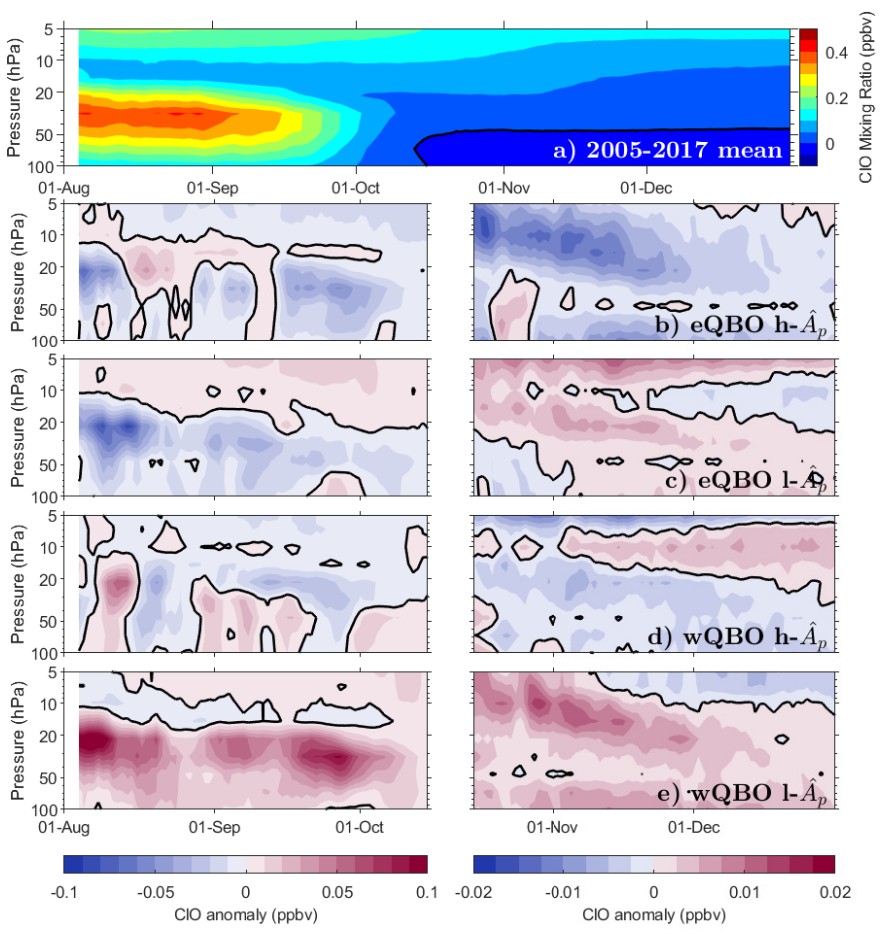

**Figure 6.** a) Polar (60°S to 82°S, area weighted) daytime MLS ClO composite mean over 2005–2017. Contour interval is 0.05 ppbv. Vertical axis is pressure from 100 hPa to 5 hPa and horizontal is time from 1 August to the end of December. Panels b) – e) anomaly from mean for b) eQBO years & high $\hat{A}_p$, c) eQBO years & low $\hat{A}_p$, d) wQBO years & high $\hat{A}_p$, and e) wQBO years & low $\hat{A}_p$. Due to the large change in ClO levels taking place in October the left column presents the anomaly from 1 August to 15 October (contour interval 0.01 ppbv) and the right column continues from 15th October to end of December (contour interval 0.0025 ppbv). Black line indicates zero contour.



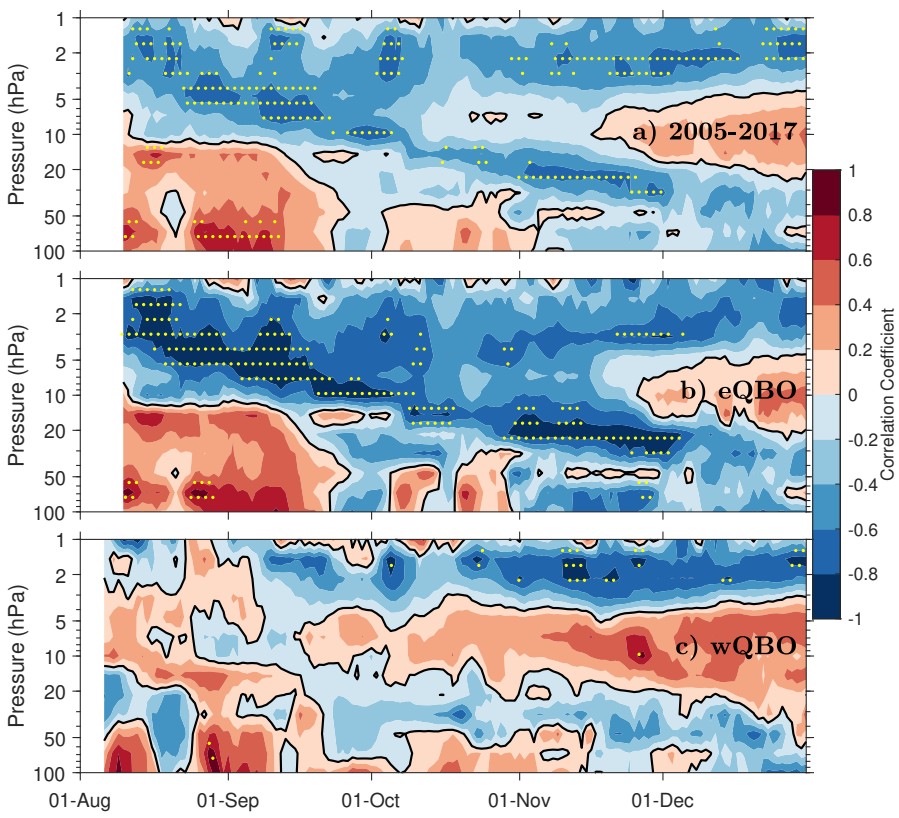

**Figure 7.** Correlation between $\hat{A}_p$ and detrended daytime MLS ClO averaged over $60°$S to $82°$S weighted by *cos(latitude)* for a) all years, b) eQBO years and c) wQBO years. Vertical axis is pressure from 100 hPa to 1 hPa and horizontal is time from 1 August to the end of December. Colour contours show correlation coefficient with contour interval 0.2 and black line indicates zero contour. Stippling indicates statistical significance ($p \leq 0.05$).

does not have the same significant anti-correlation descending in the stratosphere, but does show a weak negative correlation following approximately the same descent pattern. We note there is also a significant anti-correlation in the upper stratosphere in November to December, also present in a) and b). This may be related to the EPP-$NO_y$ that remains in the upper stratosphere (see Figure 11 of Funke et al., 2014a) while the bulk descends to lower stratosphere.

### 4.2 ClONO$_2$ observations from ACE-FTS and MIPAS

With the MLS observations providing credible evidence that stratospheric ClO is decreasing in the spring following elevated EPP during the polar winter, we now look for evidence of this being linked to enhanced levels of $NO_x$. The proposed buffering of ClO takes place via reaction (R5) which converts the ClO to ClONO$_2$. This would remove both $NO_2$ and the active $Cl_x$

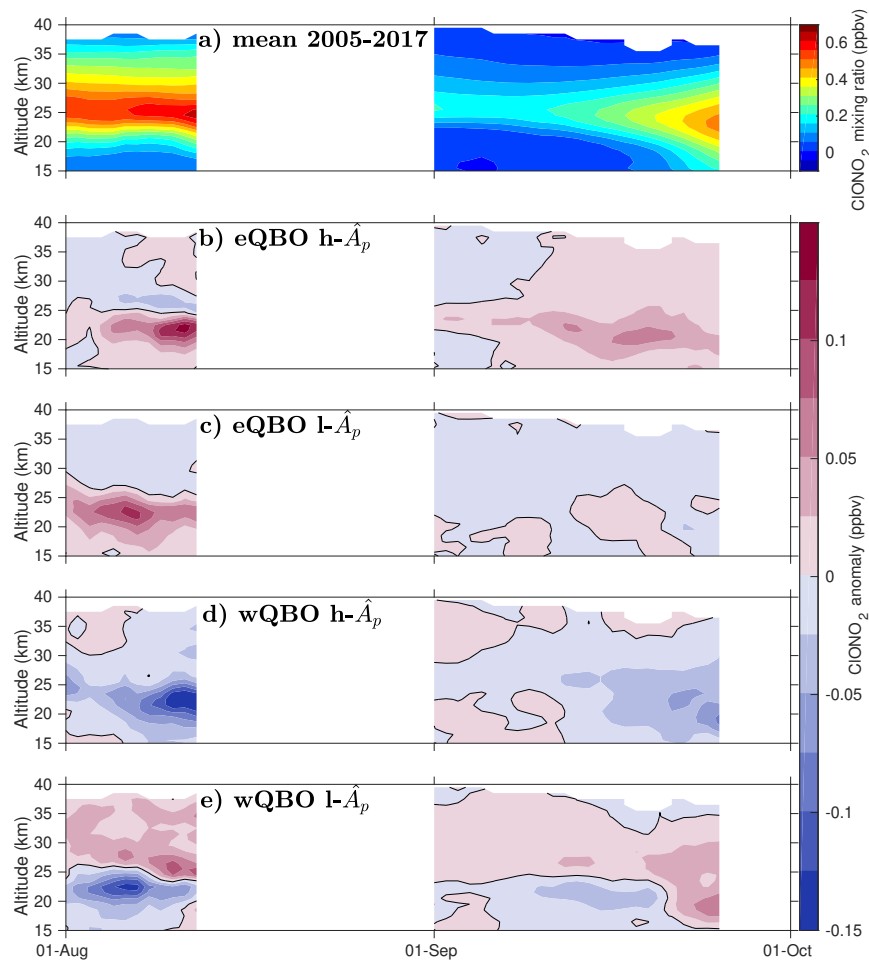

**Figure 8.** ACE-FTS $ClONO_2$: a) The composite mean for 2005 – 2017 for area weighted observations poleward of $60°$S. Horizontal axis is date from the 1st of August until the 1st of October. Vertical axis is altitude from 15 km to 40 km ($\sim$120-2 hPa). Contour interval is 0.1 ppbv. b) The anomaly from the mean for eQBO years with high $\hat{A}_p$, c) eQBO years with low $\hat{A}_p$, d) wQBO years with high $\hat{A}_p$, and e) wQBO years with low $\hat{A}_p$. Vertical axis is altitude from 15 km to 40 km. Axes as in a) with contour interval 0.05 ppbv. Black line shows zero contour.

from the catalytic ozone loss reactions, therefore resulting in overall ozone increase. To check whether $ClONO_2$ is increasing while ClO is decreasing, we analyse both ACE and MIPAS $ClONO_2$, to mitigate some of the coverage limitations of the observations.

Figure 8 presents the mean ACE-FTS $ClONO_2$, as well as the anomalies for the different QBO phase and $\hat{A}_p$ level combinations, as before. Figure 8a) displays the composite mean of a 3-d running mean $ClONO_2$ from the beginning of August to





the end of September averaged over 60°S to 90°S (weighted by *cos(latitude)*). The vertical scale here is altitude from 15 km to 40 km (~120-2 hPa). The orbit of ACE is designed to provide latitude patterns that repeat each year, allowing comparison between years as yearly coverage is approximately the same. For the ACE-FTS and MIPAS analysis, we only include days where observations were recorded within 60°S to 90°S each year of the study. As the second half of August is not consistently observed by ACE every year, these measurements were not included. Note that ACE observations from August are taken at sunrise, and September observations are from sunset. Panel a) highlights the large diurnal variation in $ClONO_2$: $ClONO_2$ is photolysed by UV radiation and thus there is more in the atmosphere at sunrise than at sunset times (i.e. higher maximum in August than in September). The minimum that occurs below 25 km around the beginning of September is due to heterogeneous chemistry destroying $ClONO_2$ on the surface of PSCs (Brasseur and Solomon, 2005) while also inhibiting $ClONO_2$ formation as PSCs remove $NO_2$ via denitrification in the lower stratosphere. $ClONO_2$ recovers around the time PSCs begin to disappear in late September. The anomalies for the combinations of $\hat{A}_p$ and QBO phase (as shown in previous figures) are shown in panels b)-e). The anomaly is variable in all cases, except for the lower stratosphere in August, which shows positive anomaly in August of the eQBO years (b and c) and negative anomaly in wQBO years (d and e). The anomalies in September are much smaller and mainly appear to show patterns in the lower stratosphere in mid to late September, once again showing positive anomaly in eQBO years and negative anomaly in wQBO years.

The altitude resolved correlation between ACE-FTS $ClONO_2$ and $\hat{A}_p$ is shown in Figure 9. Here, areas of consistent positive correlation ($\rho \geq 0.6$) occur in September in panel a) (all years) and panel b) (eQBO). These are statistically significant mostly in the middle and upper stratosphere in panel a), and in the lower stratosphere in panel b). Panel a) appears to support the hypothesis that ClO decreases are due to reactions forming $ClONO_2$. This is further supported by panel b) which also shows that eQBO amplifies the signal. Panel c) shows little consistent statistically significant correlation at this time.

Due to the limited coverage in the spring, it is difficult to draw conclusive statements from ACE-FTS observations alone. Thus we also analyse MIPAS $ClONO_2$ observations. Figure 10a) presents the mean of $ClONO_2$ from MIPAS (we only use years 2006 – 2011 here) averaged over 60°S to 90°S, weighted by *cos(latitude)*. Due to the relatively small number of years available, we only include regions that are observed every year, hence white regions correspond to places that have missing coverage at some point from 2006-2011. Here we see that $ClONO_2$ decreases throughout November in the lower stratosphere, below 30 km. Panels b) and c) show the anomaly for the composite mean of high $\hat{A}_p$ years and low $\hat{A}_p$, respectively. Note that as the time series is different, the designation of high and low $\hat{A}_p$ changes slightly: the mean $\hat{A}_p$ for 2005 – 2011 is 6.6, and we take this as limit for low and high $\hat{A}_p$. As this time period is shorter than that of OMI, MLS and ACE-FTS, we do not sort for QBO here. Years with high $\hat{A}_p$ (panel b) show consistent positive anomaly (up to +0.06 ppbv) in the middle to upper stratosphere in early September, with this positive anomaly appearing to descend to around 23 km by late November–early December. This anomaly in late spring is consistent with the altitude range where we find ozone increasing with high $\hat{A}_p$ (Figure 2), although below ~20 km, the anomaly is negative. Similarly we find descending negative anomaly in low $\hat{A}_p$ years (up to −0.06 ppbv). These results support the hypothesis that the $O_3$ increases in high $\hat{A}_p$ years result from enhanced $NO_2$ driving ClO to its $ClONO_2$ reservoir.

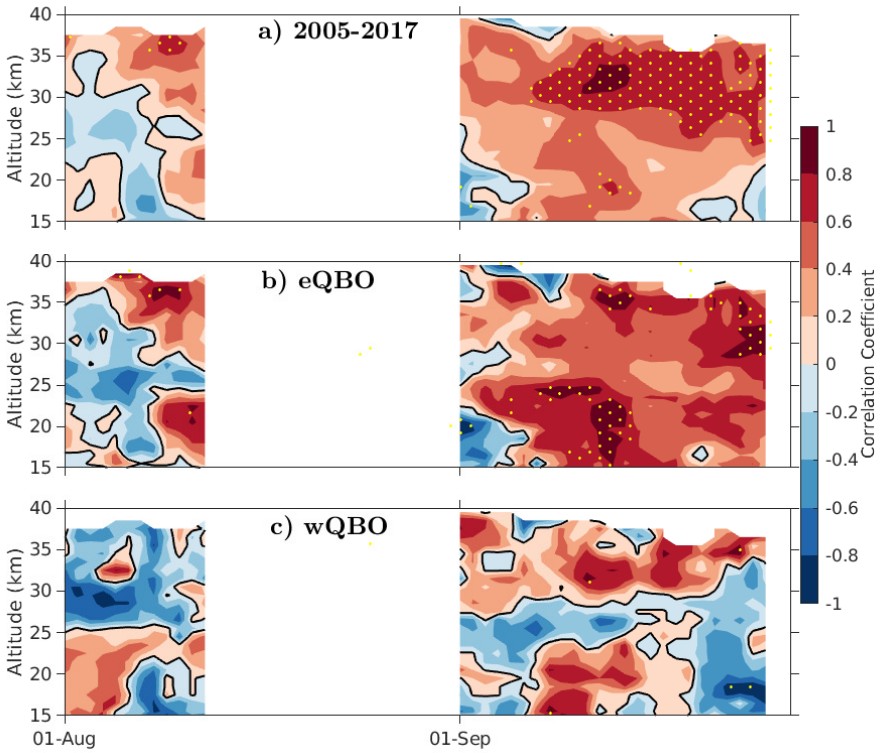

**Figure 9.** Correlation between $\hat{A}_p$ and area weighted ACE-FTS $ClONO_2$ poleward of $60°$S for a) all years, b) eQBO years, and c) wQBO years. Vertical axis is altitude from 15 km to 40 km and horizontal axis is time from 1st August to end of September. Colour contour show correlation coefficient with contour interval 0.2 and black line for 0 contour. Stippling indicates statistical significance.

The altitude correlation between $\hat{A}_p$ and MIPAS $ClONO_2$ is shown in Figure 11. We again see a descending feature similar

to those in Figures 2 and 7. As this feature shows positive (often significant) correlation ($\rho > 0.6$) it is likely that this again is due to descending EPP-$NO_x$. Note also that as the $ClONO_2$ increases appear to coincide with ClO decreases, it is unlikely that this correlation is due to the decrease in EESC over this time period as that would result in each correlation being the same sign. This figure shows that more $ClONO_2$ forms in high $\hat{A}_p$ years, and in the same area as ClO decreases (Figure 7), implying that the ClO depletion found earlier is due to $ClONO_2$ formation.

## 5    Conclusions

We have presented observational evidence that Antarctic springtime stratospheric ozone increases are associated with higher than average EPP during the preceding winter. Ozone increases due to the so called EPP indirect effect had been previously suggested (Funke et al., 2014a), but, to our knowledge, this is the first time this has been shown in observations. Following





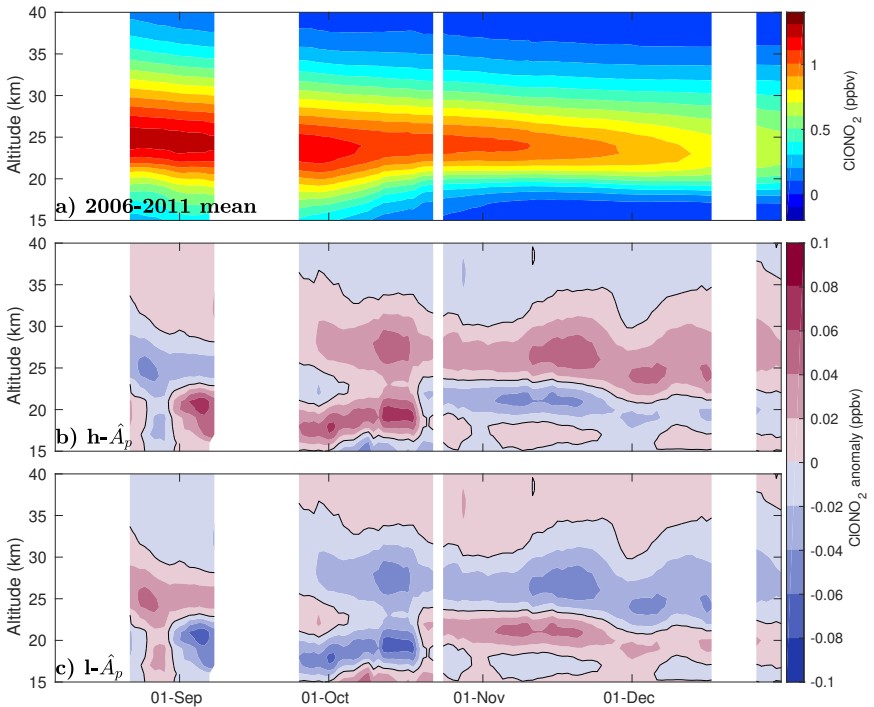

**Figure 10.** a) Composite mean of area weighted MIPAS ClONO$_2$ averaged over 60°S to 90°S from 2006 to 2011. Time is early August until the end of December, and the altitude range is 15 km to 40 km (∼120-2 hPa). Contour interval is 0.1 ppbv b) anomaly for the composite mean of years with high $\hat{A}_p$. Axes as above with contour interval 0.01 ppbv. c) as b) but for years with low $\hat{A}_p$.

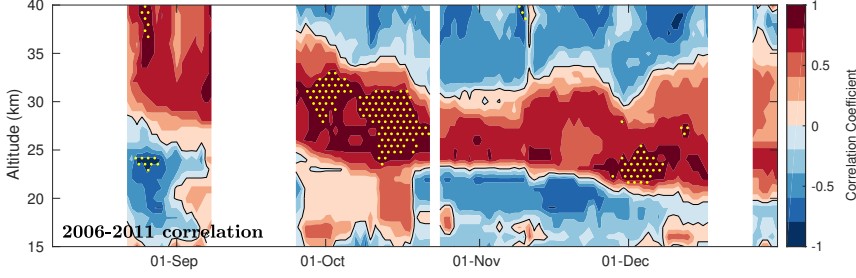

**Figure 11.** Correlation between $\hat{A}_p$ and area weighted polar MIPAS ClONO$_2$ for 2006 − 2011. Horizontal axis is date from early August until the end of December while vertical axis is altitude from 15 km to 40 km. Colour contours indicate correlation coefficient with contour interval 0.2 and black contour show the zero contour. Stippling indicates statistical significance.





the results of Gordon et al. (2020), we propose that this is due to EPP-$NO_x$ which remains the lower stratosphere at least until

November, having originally been transported from the mesosphere within the polar vortex. Jackman et al. (2000) and Funke et al. (2014a) further proposed that should this $NO_x$ reach the lower stratosphere (as shown by Gordon et al., 2020), it would react with ClO to form $ClONO_2$, preventing some of the $NO_x$ *and* $Cl_x$ driven catalytic ozone destruction. We examined polar ClO and $ClONO_2$ during the Antarctic spring and found decreases in ClO with consistent increases in $ClONO_2$ associated with above average EPP. Thus, this provides direct observational evidence supporting the hypothesis of Jackman et al. (2000);

Funke et al. (2014a).

Throughout the analysis (where possible) we have controlled for the phase of the QBO. Gordon et al. (2020) suggested that the QBO affects $NO_x$ in the stratosphere via its influence on both transport of trace gases from the equatorial region, and on wave forcing in the polar region (i.e. the Holton-Tan effect). Here, we have again seen the importance of the QBO: correlations of ozone with $\hat{A}_p$ are higher ($\geq 0.6$ in OMI total ozone) and with more occurrences of statistical significance in eQBO years.

This is in agreement with the higher correlation found in eQBO years between $NO_2$ and $\hat{A}_p$ by Gordon et al. (2020). Our results further underline the appreciable effect of the QBO on the lower polar springtime stratosphere, and that the QBO phase should be accounted for in long-term studies of this region.

Our results have shown that the EPP indirect effect has indeed affected ozone over the period 2005-2017, likely due to the interference of EPP-$NO_x$ in $Cl_x$ catalysed ozone destruction. This period has also been marked by the continuing formation

of the ozone hole every spring, although following the Montreal Protocol, the size of the ozone hole is generally decreasing with time (Solomon et al., 2016). The mechanism suggested in this paper ($NO_2$ buffering ClO) requires chlorine activation in the spring, but as chlorine loading in the polar stratosphere continues to decrease with the ban in CFC emissions, EPP-$NO_2$ will no longer hinder ozone depletion, likely instead becoming a major contributor. As ozone itself plays a vital role in both atmospheric chemistry and dynamics, this reinforces the importance of accounting for EPP in predicting the future of the polar

middle atmosphere.

*Data availability.* All data used here are open access and available from the following sources: A$_p$: http://wdc.kugi.kyoto-u.ac.jp/kp (last access: 22 January 2019); QBO: https://www.geo.fu-berlin.de/en/met/ag/strat/produkte/qbo (last access: 27 December 2019); OMI and MLS: https://earthdata.nasa.gov (last access: 24 July 2019); ACE-FTS: http://www.ace.uwaterloo.ca (registration required, last access: 23 May 2019), MIPAS: The IMK/IAA MIPAS product is available directly from IAA, IMK or https://www.imk-asf.kit.edu/english/308.php (last

access: 20 June 2020)

*Author contributions.* EMG and AS planned the study with analysis performed by EMG. EMG and AS prepared the manuscript with comments from all authors. BF provided the IMK/IAA MIPAS observations, processed the data and provided expertise on use of MIPAS data. JT provided expertise on use of OMI observations. KAW provided the expertise on use of ACE-FTS observations.





*Competing interests.* The authors declare that they have no competing interests.

*Acknowledgements.* E M Gordon was supported by a University of Otago postgraduate publishing bursary. The Atmospheric Chemistry Experiment (ACE), also known as SCISAT, is a Canadian-led mission mainly supported by the Canadian Space Agency. We acknowledge the World Data Center for Geomagnetism, the Freie Universität Berlin for the Ap and QBO data respectively. We are also grateful to the National Aeronautics and Space Administration, Canadian Space Agency and European Space Agency for providing and maintaining the high quality, long term satellite observations used in this study.



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
