# Peer review of "Observational evidence of EPP- $NO_x$ interaction with chlorine curbing Antarctic ozone loss"

_Atmospheric Chemistry and Physics, 2020_

## Referee Comment (RC1) · Anonymous Referee #1 · 30 Oct 2020

The article presents observational evidence of an indirect buffering effect of EPP on the Antarctic ozone loss in spring, by examining multiple satellite datasets. The observations reveal increases in ozone during high geomagnetic activity anomalies, which descend over the spring, and these anomalies are modulated by the QBO, such that they are only observed in years when equatorial winds at 25 hPa are easterly. Corresponding anomalies in ClO and ClONO2 support the mechanism proposed. The paper is clearly written and the results are of high interest. I only have minor comments that should be addressed before publication.

General comments

- Section 3.1. The discussion of Figure 2 does not highlight a feature that seems quite outstanding to me: there is a dipole of descending anomalies (negative above positive) in ozone linked to high Ap and easterly QBO. Does this dipole imply an upward displacement of the region of ozone loss in the presence of high Ap? The focus is on the positive anomalies in the lower stratosphere in November, which results in strong column ozone anomalies. But I think it is worth highlighting and interpreting the negative anomalies above. Also, the anomalies in Fig. 1 are described one by one but a comprehensive view is missing. For instance, the anomalies in November and December are examined separately but they clearly show a continued pattern, highlighting the mentioned dipole. This pattern linked to Ap is also seen in August-September, and disappears in October when the signal is dominated by the QBO.

- Fig. 5A: There is a clear outlier in the wQBO points, with a very low polar O3 value. Does this influence the results? It is mentioned in Section that some years corresponding to rare extreme events are not considered. Is the polar vortex that winter extremely strong or long-lasting? Should this year not be considered?

Specific comments

- L165: It is not specified in which month you select the sign of the QBO. In Table 1 it is stated that it is May. Why pick the sign of the QBO in May, when your analyses focus on August-December?

Technical

- Introduce what is Ap in abstract (L6) - L110 : remove ',' ? - L122 (also other places throughout the text, L181, 264, 266, Fig. 5a) : gradient of the trend → this expression is confusing, it should be the slope of the regression, or simply the trend - L203: 're-duction' should be 'increase', if I understand correctly - L311: 'won't' → will not - L372: 'and' is in italic format

---

## Referee Comment (RC2) · Mark Weber (Referee) · 14 Dec 2020

**1 Summary**

This paper reports on the connection between energetic particle precipitation (EPP) and various trace gases like chlorine substances (ClO and ClONO$_2$) and ozone during Antarctic winter/spring season. Their main finding is that ozone increases with elevated EPP, which is on first sight in contrast to expected decreases mainly due to enhanced NO$_x$ levels directly destroying ozone. The explanation for the ozone increases is the reduction in active chlorine by additional conversion into reservoir species due to elevated NO$_2$ that inhibits additional ozone loss. This is, however, only statistically significant in winter/spring seasons during QBO east phases.

[Figure]

The analysis methods and results are well described in this study and publication is highly recommended. Nevertheless the papers lacks detailed discussion on the interpretation of the findings.

**2 More discussion needed**

I miss in this paper a discussion on the possible reasons why the correlation with $A_p$ (the proxy for EPP) are only statistically significant during eastern QBO phase. In a brief statement the authors refer to the Holton-Tan mechanism (l. 376ff) but do not elucidate further on it. No explicit explanation is given why eQBO and not wQBO shows more significant result.

An important driver for polar ozone losses are stratospheric temperatures being sufficiently low. eQBO phases favors planetary wave propagation to be directed towards higher latitudes (see e.g. Baldwin et al. 2001) thus leading to higher stratospheric temperatures, higher ozone (NOy) transport and weaker polar vortices and less polar ozone loss. Consequently more ozone and $NO_x$ (less denitrification) are then available (see for instance Sonkaew et al., doi: doi:10.5194/acp-13-1809-2013, and references therein). The warmer the polar stratosphere the stronger the diabatic descent inside the polar vortex becomes which makes the downward transport of EPP $NO_x$ possibly more efficient during eQBO. So this could be potential mechanism that could explain the better statistics during eQBO.

Another point is that most of the (anti-)-correlation between $A_p$ and the trace gases investigated show the highest statistical significance mostly in the upper (late winter) and middle stratosphere (spring) which is above the lower stratosphere where most of the polar ozone loss occurs. This would suggest that polar ozone loss may be less affected by EPP, but the dissolution of the ozone hole over late spring may be accelerated by a faster back conversion of active chlorine into their reservoirs due to

excess $NO_2$ from EPP.

I think these points need be addressed in more detail in this paper.

**3 Minor issues**

line 5: omit "overall"

line 21: here you have a comma/semi-colon separated list, so each item should not start with capital letters, i.e. "the Brewer-Dobson circulation ...; the strong polar vortex ...; polar stratospheric clouds ..."

line 26ff: the phrase on PSC and $Cl_x$ catalytic cycle is muddied. first: PSCs convert reservoir gases into active chlorine (mainly $C_2$), the sun then activates $Cl_x$ from photolysis of $Cl_2$. The breakdown of CFCs (into reservoir gases) is mainly occurring outside the Antarctic vortex. Reaction R1 and R2 are not the main reactions in the lower stratosphere (mainly due to lack of atomic oxygen), so here the role of the ClO dimer is more relevant here.

l. 140: "anomaly study" –> "anomalies"

l. 141: line 149 "We exclude 2002 due to the sudden stratospheric warming that occurred in the SH that spring, disrupting the polar stratosphere therefore any $NO_x$ descent." During that winter there were particularly high amounts of $NO_x$ available and also strongly descended as in other winters, so there is not necessarily a disrupted $NO_x$ descent. I suggest to make a more general statement that winter/spring seasons with abrupt surges in EPP in the middle of the winter/spring (Halloween 2013) and other perturbances that lead to sudden changes in or in-situ production of $NO_x$ in the course of the winter seasons (like major warmings) were excluded from this study and that the focus is here on $NO_x$ from EPP coming from higher altitudes and descending into the stratosphere over the winter season.

l. 153: "EPP effects from the previous winter". Does that mean that the $A_p$ average from May to August (Section 2.5) is a proxy for EPP a year before. Please clarify.

Table 1: suggest to mention in the table caption the delimiter value which separates low and high $A_p$ values.

line 210: "As in Figure 1, ozone is cos(latitude) weighted zonal mean average over 60S to 82S. Note that for all correlation analyses presented here, the data has been linearly detrended to avoid misattribution from linear increases or 215 decreases from reduced EESC since 2005." This has been already stated before and does not need to be repeated here again.

Figure 4: Why is there a data gap in OMI near October 1. By averaging many years there should be no gaps.

Figure 5: In panel (a) there are two data points from wQBO that rather fit to eQBO and one from eQBO to wQBO regression line. Some comments on that. Are there winter/spring seasons with QBO phase changes in the middle of the season? Can they cause outliers? What about years where $A_p$ changes strongly from May to August?

Figure 5: "Recall eQBO years are [2005 2007 2009 2010 2012 2014 2017]." I would rather refer to Table 1 and omit this.
* * *

---

## Author Comment (AC1) · 22 Dec 2020

We would like to thank both reviewers for their comments. Our detailed responses to all comments are included here.

**Comments from Anonymous Referee (RC1)**

**General comments:**

Comment: Section 3.1. The discussion of Figure 2 does not highlight a feature that seems quite outstanding to me: there is a dipole of descending anomalies (negative above positive) in ozone linked to high Ap and easterly QBO. Does this dipole imply an upward displacement of the region of ozone loss in the presence of

high Ap? The focus is on the positive anomalies in the lower stratosphere in November, which results in strong column ozone anomalies. But I think it is worth highlighting and interpreting the negative anomalies above. Also, the anomalies in Fig. 1 are described one by one but a comprehensive view is missing. For instance, the anomalies in November and December are examined separately but they clearly show a continued pattern, highlighting the mentioned dipole. This pattern linked to Ap is also seen in August-September, and disappears in October when the signal is dominated by the QBO.

Reply: The negative correlation pattern is likely closely linked with the known EPP-$NO_x$ descent pattern and we have added more emphasis on this to the text. The dipole patterns is very interesting and similar results have been reported from previous model studies by Andersson et al. 82018). However, the positive anomalies have little statistical significance before November, hence our earlier focus on the November pattern.

We have revised the text to emphasise the descending negative pattern in Figure 2: *... descending in the polar vortex, as the pattern of descending significant negative correlation is consistent with the reported descending EPP-$NO_x$ "tongue" (see e.g. Funke et. al 2014a).* and discuss the dipole pattern more: *We note that the positive correlation pattern does appear earlier and seems to descend with the negative pattern, but the positive correlation does not become statistically significant until November. A similar dipole pattern has previously been seen in model simulations with suggestions that it may be linked to chlorine and bromine chemistry (Jackman et al., 2009; Andersson et al., 2018). Our results here seem ...*

We also revised the text to tie together the main points from Figure 1: *Overall, Figure 1 provides evidence of the combined role of the QBO and EPP on ozone in the Antarctic stratosphere, with $\hat{A}_p$ important in the mid to upper stratosphere in early spring however the QBO tends to dominate in the lower stratosphere*

*in mid-Spring (positive anomaly with eQBO, negative anomaly for wQBO) and EPP appearing to affect the signal in the lower stratosphere in mid November (negative for high $\hat{A}_p$, positive for low $\hat{A}_p$).*

Comment: Fig. 5A: There is a clear outlier in the wQBO points, with a very low polar O3 value. Does this influence the results? It is mentioned in Section that some years corresponding to rare extreme events are not considered. Is the polar vortex that winter extremely strong or long-lasting? Should this year not be considered?

Reply: This point corresponds to year 2015, when the area of Antarctic ozone hole was one of the largest over observed. Solomon et al. (Science, 2016) have partially attributed the 2015 ozone loss to a volcanic eruption, as well as interannual variability. We have excluded years from the earlier MIPAS observations when the upper atmospheric $NO_x$ source is expected to be anomalous (e.g. due to SSW) and this is addressed in the section about MIPAS data. However, this case did not unambiguously influence the $NO_x$ source so we included it in the analysis, but have now added a note stating that the ozone hole was particularly large that year: *Note that the wQBO year with detrended polar ozone less than 50 DU corresponds to the year 2015, when the ozone hole has been reported to be particularly large in area (Solomon et al., 2016).*

**Specific comments:**

Comment: L165: It is not specified in which month you select the sign of the QBO. In Table 1 it is stated that it is May. Why pick the sign of the QBO in May, when your analyses focus on August-December?

Reply: We chose May as this is when QBO is though to affect vortex formation, and thus $NO_x$ descent conditions in winter. We provide detailed discussion and context, with references, on this in Gordon et. al (2020) which the current work builds

on. We have now clarified this in the text and have added a further reference to the previous work: *...according to the phase of the QBO in May as QBO in this month captures the effect of the QBO on the polar vortex (see Gordon et. al 2020)*

**Technical:**

Comment: Introduce what is Ap in abstract (L6)

Reply: We corrected this to: *Using the geomagnetic activity index $A_p$ to proxy EPP...*

Comment: L110 : remove ','?

Reply: Removed.

Comment: L122 (also other places throughout the text, L181, 264, 266, Fig. 5a) : gradient of the trend → this expression is confusing, it should be the slope of the regression, or simply the trend

Reply: These have been changed accordingly:

– overall *slope* of the trend
– calculating the *slope* of the yearly trend
– Figure 5 now shows the *slope of the regression* between
– We find that the *slope* is positive

Comment: L203: 'reduction' should be 'increase', if I understand correctly

Reply: This is correct. We have revised the text as suggested.

Comment: L311: 'won't' → will not

Reply: This has been revised as suggested.

Comment: L372: 'and' is in italic format

Reply: This has been changed to normal format.

**Comments from Mark Weber (RC2)**

Comment: I miss in this paper a discussion on the possible reasons why the correlation with Ap (the proxy for EPP) are only statistically significant during eastern QBO phase. In a brief statement the authors refer to the Holton-Tan mechanism (l. 376ff) but do not elucidate further on it. No explicit explanation is given why eQBO and not wQBO shows more significant result.

Reply: We did not initially include this discussion as it was addressed in detail in Gordon et al., 2020 which this work builds on. Following the comment we realise that this is indeed needed here and have added more information to the introduction as well as conclusions.

In the introduction we now write: [Gordon et. al] *show evidence that the QBO affects the temperature of the polar vortex in winter with warmer vortex in easterly QBO (eQBO) years. This leads to inhibited PSC formation and hence less effective removal of nitrogen species from the lower stratosphere.*

We now write in the conclusions: *As Gordon et. al (2020) proposed in the context of Antarctic $NO_2$ column, we suggest that the reasons for the QBO modulating Antarctic ozone loss are also via its effect on wave-forcing in the polar region (i.e. the Holton-Tan effect). Gordon et. al (2020) showed that eQBO years were more likely to have a warmer Antarctic vortex and proposed that this would lead to less denitrification in the lower stratosphere, resulting in a less suitable environment for PSC formation. As PSCs are crucial to springtime ozone loss in the*

*lower stratosphere in springtime, we suggest that the inhibited PSC formation in eQBO years contributes to our findings that less chlorine is activated from reservoirs, and hence less ozone loss in eQBO years, with EPP-NO$_x$ contributing to increased ClONO$_2$ formation (see R5). This is similar to Sonkaew et. al (2013), who for the Northern Hemisphere found that years with a warmer Arctic vortex resulted in less springtime ozone loss. We suggest occurs this in the Southern Hemisphere, and but also reinforce the important role played by EPP-NO$_x$.*

Comment: An important driver for polar ozone losses are stratospheric temperatures being sufficiently low. eQBO phases favors planetary wave propagation to be directed towards higher latitudes (see e.g. Baldwin et al. 2001) thus leading to higher stratospheric temperatures, higher ozone (NO$_y$) transport and weaker polar vortices and less polar ozone loss. Consequently more ozone and NO$_x$ (less denitrification) are then available (see for instance Sonkaew et al., doi: doi:10.5194/acp-13-1809-2013, and references therein). The warmer the polar stratosphere the stronger the diabatic descent inside the polar vortex becomes which makes the downward transport of EPP NO$_x$ possibly more efficient during eQBO. So this could be potential mechanism that could explain the better statistics during eQBO.

Reply: We agree and have addressed this above in our addition to the conclusion where we also now include a reference to Sonkaew et al. 2013.

Comment: Another point is that most of the (anti-)-correlation between Ap and the trace gases investigated show the highest statistical significance mostly in the upper (late winter) and middle stratosphere (spring) which is above the lower stratosphere where most of the polar ozone loss occurs. This would suggest that polar ozone loss may be less affected by EPP, but the dissolution of the ozone hole over late spring may be accelerated by a faster back conversion of active chlorine into their reservoirs due to excess NO$_2$ from EPP. I think these points need

be addressed in more detail in this paper.

Reply: We have added more detailed discussion on the descent patterns and their significance (please see reply to RC1). We also added to the text to discuss the descent pattern in the conclusions (i.e. the reason for the correlation pattern begins in the upper stratosphere in winter etc): *We were able to trace this descent pattern in observations of $O_3$, ClO and $ClONO_2$, finding it matched that of the previously reported descent of EPP-$NO_x$ see e.g Randall et. al (2006).*

We further revised the text to emphasise our speculated role of $NO_x$ in slowing down ozone loss: *Thus, this provides direct observational evidence supporting the hypothesis of ... that ozone loss may be decelerated in the Antarctic lower stratosphere following winters with high EPP years due to excess $NO_x$ accelerating ClO back to its reservoirs.*

We also added the explicit comment on the role of the lower stratospheric processed to summarize the chlorine section: *Overall, these results suggest that the arrival of EPP-$NO_x$ in the lower stratosphere by the late Antarctic springtime is contributing to faster conversion of active chlorine into reservoir species, which could bring about the end of the springtime ozone hole faster (as seen in the enhanced OMI total column ozone).*

**Minor issues:**

Comment: line 5: omit "overall"

Reply: This word was omitted as suggested.

Comment: line 21: here you have a comma/semi-colon separated list, so each item should not start with capital letters, i.e. "the Brewer-Dobson circulation ...; the strong polar vortex ...; polar stratospheric clouds ..."

Reply: Thank you for pointing this out, we have now corrected this.

Comment: line 26ff: the phrase on PSC and Clx catalytic cycle is muddied. first: PSCs convert reservoir gases into active chlorine (mainly C2), the sun then activates Clx from photolysis of Cl2. The breakdown of CFCs (into reservoir gases) is mainly occurring outside the Antarctic vortex. Reaction R1 and R2 are not the main reactions in the lower stratosphere (mainly due to lack of atomic oxygen), so here the role of the ClO dimer is more relevant here.

Reply: Thank you, we have removed mention of CFCs broken down in PSCs. We have also de-emphasised R1 and R2:

*$Cl_x$ is effective at catalytically destroying ozone, with one such chain of reactions:*

and now mention the role of the dimer cycle: *Other, more complicated reactions such as with ClO dimer, and heterogeneous reactions also destroy ozone..., but they will not be elaborated on further here.*

Comment: l. 140: "anomaly study" → "anomalies"

Reply: This has been revised as suggested.

Comment: l. 141: line 149 "We exclude 2002 due to the sudden stratospheric warming that occurred in the SH that spring, disrupting the polar stratosphere therefore any NOx descent." During that winter there were particularly high amounts of NOx available and also strongly descended as in other winters, so there is not necessarily a disrupted NOx descent. I suggest to make a more general statement that winter/spring seasons with abrupt surges in EPP in the middle of the winter/spring (Halloween 2013) and other perturbances that lead to sudden changes in or in-situ production of NOx in the course of the winter seasons (like major warmings) were excluded from this study and that the focus is here on NOx from EPP coming from higher altitudes and descending into the stratosphere over the winter season.

Reply: We have revised this accordingly and now write in more general terms here: *For our analysis, we exclude observations from the years before the instrument error due to events that resulted in surges of NO$_x$ in the stratosphere due to transport or in situ production during the SH winter/spring (López-Puertas et al., 2005; Funke et al., 2014), and utilise MIPAS ClONO$_2$ observations...*

Comment: l. 153: "EPP effects from the previous winter". Does that mean that the Ap average from May to August (Section 2.5) is a proxy for EPP a year before. Please clarify.

Reply: By previous winter we mean the preceding winter in the Southern Hemisphere, i.e. in the same calendar year (winter season is JJA, spring is SON). We have now clarified this in the text.

Comment: Table 1: suggest to mention in the table caption the delimiter value which separates low and high Ap values.

Reply: We have added information on the delimiter value of 8.3 to the caption as requested.

Comment: line 210: "As in Figure 1, ozone is cos(latitude) weighted zonal mean average over 60S to 82S. Note that for all correlation analyses presented here, the data has been linearly detrended to avoid misattribution from linear increases or 215 decreases from reduced EESC since 2005." This has been already stated before and does not need to be repeated here again.

Reply: We removed this text as requested.

Comment: Figure 4: Why is there a data gap in OMI near October 1. By averaging many years there should be no gaps.

Reply: This gap is a result of the correlation method used; when calculating the correlation coefficient, if the input has a missing value, then the output is a missing value for the correlation coefficient. We have amended the text: *Note the missing values in late September are due to missing values in the time series. We have chosen not to calculate the correlation coefficient for these points so as not to be misleading about the number of years in each correlation calculation.*

Comment: Figure 5: In panel (a) there are two data points from wQBO that rather fit to eQBO and one from eQBO to wQBO regression line. Some comments on that. Are there winter/spring seasons with QBO phase changes in the middle of the season? Can they cause outliers? What about years where Ap changes strongly from May to August?

Reply: The two wQBO years are 2013 and 2016. In 2013 there are no phase changes during the year. In early 2016 the widely documented wQBO disruption occurred in February, this is well before the SH early winter period, but it is possible there are downstream effects from the disrupted dynamics. The outlier eQBO year corresponds to 2010 when the phase does change later on during the winter. We have added more discussion on these, as well as a comment on case of wQBO with very low detrended polar ozone year of 2015.

The text regarding Figure 5 now includes the following discussion: *Note that the wQBO year with detrended polar ozone less than 50 DU corresponds to the year 2015, when the ozone hole has been reported to be particularly large in area (Solomon et al. 2016). The two wQBO years with the highest detrended ozone columns correspond to years 2013 and 2016, the latter of which presented a disruption in the QBO phase in February (Newman et al., 2016). The eQBO year with lowest ozone column corresponds to the year 2010. The QBP phase in 2010 changed during the Antarctic winter season from eQBO to wQBO, and this may have contributed to the low polar ozone amount in November.*

[Figure]

Abrupt changes in Ap are possible, but previous work (including Siskind et al. 2000, Seppälä et al., 2007, Funke et al., 2014a, Gordon et al., 2020, and others) has shown that an averaged Ap provides a reasonably good proxy for the cumulative effect of EPP-NO$_x$ production above the stratopause and the following transport into the stratosphere. Once in the stratosphere, large scale dynamics appear to play another key role in understanding the year-to-year variability in the EPP-NO$_x$ reaching below 20-30 km, at least in the SH, as we found in Gordon et al. 2020. While our approach here is more statistical, follow up research on the individual years may bring to light which factors played contributed to the polar ozone variability in these particular years.

Comment: Figure 5: "Recall eQBO years are [2005 2007 2009 2010 2012 2014 2017]." I would rather refer to Table 1 and omit this.

Reply: We revised the figure caption by referring to Table 1 as suggestion and it now reads: *eQBO and wQBO years as given in Table 1*.

---

## Author Response (AR2)

Author responses to editor comments on paper "Observational evidence of EPP-NO $_x$  interaction with chlorine curbing Antarctic ozone loss".

We would like to thank the editor for his comments on the paper. We have implemented all suggestions in the text.

- Comment: The Holton-Tan mechanisms explains the influence of the QBO on the strength of polar vortex via wave forcing. This is however, a non-local effect, thus not "in the polar region". I therefore suggest to rephrase the statement in the conclusions section to something like: "the QBO modulating Antarctic ozone loss are also via its wave-forcinge effect on the polar region (i.e. the Holton-Tan effect)"
  - Reply: We changed this text as suggested.
- Comment: In Figure 5, detrended polar ozone column values of less than 100 DU are shown. Please specify more clearly what this is, as certainly the November ozone column is not below 100 DU.
  - Reply: This is caused by the area weighting applied across the polar region. We added to the figure caption to explain this, and also noted that these are low when contrasted to the non-area weighted values shown in Figure 3a.
- Comment: The new sentence summarizing Figure 1 in response to rev. 1 comments is very long. I suggest to break up into at least two sentences.
  - Reply: This has now been broken into two separate sentences.
- Comment: Rev. 1 remarked that the term "gradient of the trend" should be avoided. I agree with this statement. The replacement with slope of the trend is however not better. In mathematical terms, the slope of a trend would be the second derivative. Please simply use positive or negative trend.
  - Reply: We updated this accordingly across the manuscript.
- Comment: The use of "previous winter" is still confusing, as you are referring to the same year. I suggest that the use of "respective" winter may be clearer, inparticular if you refer to November values. In case you explicitley refer to spring (and not to November), the term "proceeding winter" would also be clear.
  - Reply: We changed to the term "preceding winter" to clarify. We also added the following example to section 2.5: *Explicitly, when investigating the Antarctic atmosphere in Aug-Dec of e.g. the year 2012, we would contrast to the average*  $A_p$  *of the preceding winter: May-August of 2012.*

---

## Author Response (AR3)

Author responses to editor comments on paper "Observational evidence of EPP-NO$_x$ interaction with chlorine curbing Antarctic ozone loss".

We would like to thank the editor for his comments on the paper. We have implemented all suggestions in the text.

Comment: I have checked all of the corrections and I am happy with the changes you made. The only remanining point from my side is the y-axis in Fig. 5. Please check if the cosine averaging has been applied correctly and the values are correctly normalized.

Reply: We revised the cosine weighting and updated the figure and text accordingly. The range of values in the y-axis is now consistent with observed values.